# OpenReview forum: "Rethinking Deep Safety Alignment: Reflective Safety Alignment for Balancing Harmlessness and Helpfulness of LLMs"
_ICLR.cc/2026/Conference — ICLR 2026 Conference Withdrawn Submission_

### Official Review · Reviewer_eNep · 2025-10-30

**Soundness:** 2
**Presentation:** 2
**Contribution:** 2
**Rating:** 4
**Confidence:** 3

**Summary:**

This paper proposes ReAlign, which improves the model safety generalization and mitigates over-alignmnt by enhancing the safety response with reasoning. The method consists of two stages: Reasoning-type warmup and self-reflective reasoning process optimization.  Experiments show that such approach helps improve the model's helpfulness and harmlessness trade-off.

**Strengths:**

The second stage of Self-Reflective Reasoning process optimization is innovative and interesting. It encourages the model to do early safety reflection, which helps the model to learn the nuance in safety related prompt and also save inference cost.

**Weaknesses:**

1. The overall novelty is limited.  The idea of using reasoning to help align the model is not new. The author only compares with STAIR as a baseline, but there are other approaches using SFT [1] or RL [2] to teach LM to reason over harmful and harmless prompts to improve and mitigate over-alignment. The author also mentioned deliberate alignment from OpenAI, which also demonstrated the importance of reasoning for alignment. The major novelty lies in the second SRPO stage.

2. It is not clear how much advantage we get by using the PP-COT besides the inference cost saving.






[1] Si, S., Wang, X., Zhai, G., Navab, N., & Plank, B. (2025). Think Before Refusal : Triggering Safety Reflection in LLMs to Mitigate False Refusal Behavior. ArXiv, abs/2503.17882.
[2] Kim, T., Tajwar, F., Raghunathan, A., & Kumar, A. (2025). Reasoning as an Adaptive Defense for Safety. ArXiv, abs/2507.00971.

**Questions:**

see weakness above

---

### Official Review · Reviewer_erxr · 2025-10-31

**Soundness:** 2
**Presentation:** 3
**Contribution:** 2
**Rating:** 4
**Confidence:** 3

**Summary:**

The paper proposes ReAlign, a reflective safety alignment framework that achieves deep safety alignment in large language models by balancing harmlessness and helpfulness. It addresses two key issues in existing alignment methods—under-generalization to unseen jailbreaks and over-alignment causing excessive refusals—arguing that current approaches (SFT, DPO, RLHF) only perform shallow token-level corrections rather than reasoning-level safety control.

**Strengths:**

ReAlign advances safety alignment from surface-level refusal to reasoning-based self-reflection, encouraging models to reason about policy violations before responding. This deepens interpretability and robustness against jailbreak attacks.

**Weaknesses:**

1. While the paper compares against STAIR and Recovery Examples, it lacks direct comparison with PPO-based alignment methods (e.g., RLHF, RLAIF). Including such baselines would better position the proposed approach within the standard alignment paradigm and strengthen the empirical claims.

2. Since reflective reasoning is explicitly embedded during training, models may learn to superficially mimic reflection patterns (e.g., "Wait, this is unsafe...") without genuinely reasoning about safety. The paper would benefit from additional analysis demonstrating that the observed reflections represent authentic reasoning rather than learned surface patterns.

3. The framework relies heavily on GPT-4o-generated reflective reasoning, which may introduce distributional biases or stylistic artifacts specific to GPT-4o. This raises concerns about generalization to other model families and real-world adversarial scenarios.

**Questions:**

1. Would reflective reasoning be integrated into PPO for further safety gains?

2. What is the actual increase in reasoning tokens and inference time compared to standard DPO?

---

### Official Review · Reviewer_Do6G · 2025-11-02

**Soundness:** 3
**Presentation:** 3
**Contribution:** 2
**Rating:** 6
**Confidence:** 4

**Summary:**

This paper introduces ReAlign, a reflective safety alignment framework for LLMs, aiming to balance harmlessness and helpfulness. The authors point out that existing alignment methods suffer from under-generalization and over-alignment. They address these issues via two stages, Reasoning-style Warmup (RW) and Self-reflective Reasoning Process Optimization (SRPO). Experiments across diverse safety and general benchmarks demonstrate that ReAlign significantly improves robustness against jailbreaks and reduces over-refusals without degrading general performance.

**Strengths:**

* The paper identifies a real trade-off between harmlessness and helpfulness in safety alignment. The problem setup is clear and reasonable.
* Authors provide thorough empirical evaluations across multiple safety benchmarks and general benchmarks, strongly supporting the effectiveness of the proposed method. The results are strong, demonstratring well-balanced performance on safety and utility.
* The paper performs a sufficient discussion on the approach with ablation studies and qualitative analyses.

**Weaknesses:**

* While the paper mentions that the “data generator” in ReAlign can be implemented by the target model itself, the experiments in practice rely heavily on GPT-4o to produce reasoning-style safety data and reflective trajectories. This raises a concern whether the satisfactory results come from high-quality data by a superior model or the method itself. There should be a study on how the method performs with other data sources or at least a discussion on this aspect.
* The paper describes its approach as “safety-policy-driven”, but the details of policy are not formally defined. It is unclear what are the policies and how they interacts with the reasoning process. Moreover, a recent work [1] explicitly introduces a policy-based safety reasoning framework, which has not been included.
* As there have been some papers studying safety alignment of Large Reasoning Models (LRMs), it is straightforward to think whether this method is applicable to LRMs. Since ReAlign itself builds upon reasoning-based safety alignment, it would be natural and valuable to analyze its relationship with such works to better define the position and contributions of this paper. Some relevant papers [2,3,4,5] are listed below.

[1] ARMOR: Aligning Secure and Safe Large Language Models via Meticulous Reasoning

[2] SafeChain: Safety of Language Models with Long Chain-of-Thought Reasoning Capabilities

[3] STAR-1: Safer Alignment of Reasoning LLMs with 1K Data

[4] RealSafe-R1: Safety-Aligned DeepSeek-R1 without Compromising Reasoning Capability

[5] SafeKey: Amplifying Aha-Moment Insights for Safety Reasoning

**Questions:**

See weaknesses.

---

### Official Review · Reviewer_ezrr · 2025-11-02

**Soundness:** 2
**Presentation:** 2
**Contribution:** 2
**Rating:** 2
**Confidence:** 5

**Summary:**

This paper proposes ReAlign, a reflective safety alignment framework for LLMs that aims to balance harmlessness and helpfulness. The approach consists of two stages: (1) Reasoning-style Warmup (RW) that enables LLMs to internalize long-chain reasoning capability, and (2) Self-reflective Reasoning Process Optimization (SRPO) that promotes reflection and correction during reasoning. The authors also claim their method reduces over-refusal while maintaining safety performance across multiple benchmarks.

**Strengths:**

- The paper addresses the practical problem of balancing safety and helpfulness in LLM alignment.

- The probe-based analysis provides some insights into model behavior during alignment.

- The writing is generally clear and the experiments cover multiple benchmarks.

**Weaknesses:**

- **Limited Core Novelty.** The core contribution of this work is limited to data construction and reformatting. The two-stage training process (reasoning-style warmup + preference optimization) lacks genuine methodological innovation. Reasoning-style fine-tuning has been widely explored in recent reasoning model literature, and stepwise decomposition of reasoning chains with reflection is not a novel concept. The paper essentially applies existing techniques (SFT + DPO) on newly constructed safety-oriented reasoning data. While data construction can be valuable, the novelty threshold for a top-tier venue requires more substantial methodological contributions. The authors need to clearly articulate what is fundamentally new beyond "collecting safety reasoning data and applying standard fine-tuning methods."

- **Suboptimal Safety Performance Compared to SOTA.** Table 2 shows that ReAlign's safety performance is substantially worse than STAIR across most metrics. For example, on LLAMA3.1-8B-IT, STAIR achieves an ASR of 1.95% on WildJailbreak, while ReAlign achieves 4.95%, more than double the attack success rate. Similarly, on jailbreak attacks (SGB variants), STAIR consistently outperforms ReAlign by significant margins. While the authors argue that ReAlign achieves better balance by reducing over-refusal, the primary objective of safety alignment is to ensure safety first. A method that sacrifices substantial safety gains for marginal improvements in helpfulness or over-refusal is questionable.

- **Limited Experimental Scope.** The experimental evaluation is limited to models with small parameters. Why not evaluate on larger models, like 14B and 32B? More critically, given the recent popularity of large reasoning models and your contribution on reasoning data construction, why is there no verification on these reasoning models? Lacking these results is unconvincing.

- **Poor Figure and Table Presentation.** The presentation quality significantly detracts from the paper's professionalism. All figures and tables contain extremely small text, which is not suitable for reading.

**Questions:**

The questions are listed in the weaknesses, and this paper needs massive revisons in my belief.

---

### Note · Authors · 2025-12-03

I have read and agree with the venue's withdrawal policy on behalf of myself and my co-authors.